# Acoustic Radiation Forced Impulse of the Liver and the Spleen, Combined with Spleen Dimension and Platelet Count in New Ratio Scores, Identifies High-Risk Esophageal Varices in Well-Compensated Cirrhotic Patients

**DOI:** 10.3390/diagnostics14070685

**Published:** 2024-03-25

**Authors:** Antonio F. M. Vainieri, Elisa Brando, Antonio De Vincentis, Giulia Di Pasquale, Valentina Flagiello, Paolo Gallo, Francesca Barone, Teresa Massaro Cenere, Evelyn Di Matteo, Antonio Picardi, Giovanni Galati

**Affiliations:** 1Department of Internal Medicine, S. Giovanni Addolorata Hospital, 00184 Rome, Italy; afmvainieri@hsangiovanni.roma.it; 2Operative Research Unit of Clinical Medicine and Hepatology, Fondazione Policlinico Universitario Campus Bio-Medico, 00128 Rome, Italy; elisabrando91@gmail.com (E.B.); g.dipasquale@policlinicocampus.it (G.D.P.); v.flagiello@unicampus.it (V.F.); paolo.gallo@policlinicocampus.it (P.G.); fbarone07@gmail.com (F.B.); dimatteo.evelyn@gmail.com (E.D.M.); a.picardi@policlinicocampus.it (A.P.); 3Operative Research Unit of Internal Medicine, Fondazione Policlinico Universitario Campus Bio-Medico, 00128 Rome, Italy; a.devincentis@policlinicocampus.it; 4Research Unit of Internal Medicine, Department of Medicine and Surgery, Università Campus Bio-Medico di Roma, 00128 Rome, Italy; 5Reasearch Unit of Clinical Medicine and Hepatology, Department of Medicine and Surgery, Università Campus Bio-Medico di Roma, 00128 Rome, Italy; 6Department of Internal Medicine, Sapienza University, 00161 Rome, Italy; tmassaro21@gmail.com

**Keywords:** ARFI, acoustic radiation forced impulse, platelet count, spleen, esophageal varices

## Abstract

Acoustic radiation forced impulse (ARFI) is an integrated ultrasound method, measuring stiffness by point shear wave elastography. To evaluate the diagnostic performance of the ARFI of the liver and the spleen, combined with spleen dimension and platelet count, in predicting high-risk esophageal varices (HRVs) in cirrhotic patients, a prospective and cross-sectional study was conducted between February 2017 and February 2021. The following ratio scores were calculated based on ARFI measurements: ALSDP (ARFI Liver–Spleen Diameter-to-Platelet Ratio Score), ASSDP (ARFI Spleen–Spleen Diameter-to-Platelet Ratio Score), ASSAP (ARFI Spleen–Spleen Area-to-Platelet Ratio Score), and ALSAP (ARFI Liver–Spleen Area-to-Platelet Ratio Score). In 100 enrolled subjects, spleen ARFI, ASSDP, and ASSAP were significantly associated with HRVs in the prospective short- and long-term follow-ups and in the cross-sectional study (*p* < 0.05), while ALSDP and ALSAP were associated with HRVs only in the prospective long-term follow-up and cross-sectional study (*p*< 0.05). ASSAP was the best ARFI ratio score for HRVs at the long-term follow-up [value of area under curve (AUC) = 0.88], although all the ARFI ratio scores performed better than individual liver and spleen ARFI (AUC > 0.7). In our study, ARFI ratio scores can predict, in well-compensated cirrhotic patients, the risk of developing HVRs in short- and long-term periods.

## 1. Introduction

Portal hypertension (PH) is the main clinical syndrome of liver cirrhosis, which is responsible for most of its complications. PH is defined as a hepatic venous pressure gradient (HVPG) greater than 5 mmHg. Based on portal HVPG, patients with clinically significant portal hypertension (CSPH) showed an HVPG of ≥10 mmHg [1]. CSPH is related to an increased risk of developing gastroesophageal varices (GEVs) [2], overt clinical de-compensation, such as ascites, variceal bleeding, and encephalopathy [3], and hepatocellular carcinoma [4]. In patients with GEVs, an HVPG of >12 mmHg identifies bleeding risk, while an HVPG < 12 mmHg is associated with protection from bleeding [5]. Since the risk of GEV bleeding can be reduced with appropriate medical or endoscopic treatment, in patients with high-risk varices (HRVs) also called “varices needing to treat” (varices with red wale marks or large varices), endoscopic screening of the upper gastrointestinal tract is currently the diagnostic standard. However, a large proportion of cirrhotic patients will not develop HRVs, thus making esophagogastroduodenoscopy (EGD) a non-ideal screening test that is associated with significant costs and patient discomfort [6]. Accordingly, in recent decades, increased attention has been dedicated to identifying some accurate non-invasive tests that can rule in and rule out CSPH and HRVs, reducing or avoiding the use of invasive methods such as HVPG measurement and EGD [7]. Among laboratory data, platelet count is inversely related to PH, but when considered alone, its accuracy for CSPH does not exceed a value of area under curve (AUC) of 0.75 in the literature [8].

Likewise, although splenomegaly taken alone is sensitive, but not a specific sign of PH, the size of the spleen, combined with platelet count and liver stiffness measurement (LSM) by transient elastography (TE) performed with Fibroscan^®^, provides accurate data on the presence of CSPH/GEVs [8,9].

TE, the first highly validated method with a high adoption worldwide, developed to measure liver stiffness (results expressed in kilopascal [kPa]) and secondly spleen stiffness, exhibited a relatively low specificity in terms of GEV prediction in one meta-analysis, although the capacity of predicting CSPH was relatively high (sensitivity: 90%; specificity: 79%) [10]. However, Fibroscan^®^ exhibits a high measurement failure rate in patients with narrow intercostal spaces, high body mass index, or ascites [11]. Another limitation is that it is based on M-mode imaging without the real-time visualization of the liver parenchyma.

In contrast, new methods integrated into the ultrasound machines can measure the tissues stiffness during a normal examination, enhancing the success rate of measurements [12] and expressing the results in meter per second (m/s) or converting in kPa with the Young’s module formula [13,14]. The first highly validated among these new methods is the point shear wave elastography (pSWE) method named also acoustic radiation forced impulse (ARFI) by certain ultrasound companies.

The aim of our study was to evaluate the diagnostic value of liver and spleen ARFI, combined with spleen dimension obtained with a conventional abdominal ultrasound and platelet count in new ratio scores, for detecting and predicting HRVs in a population of cirrhotic patients with different etiologies.

## 2. Materials and Methods

We conducted a single-center prospective and cross-sectional observational study, between February 2017 and February 2021. All consecutive patients with an established diagnosis of liver cirrhosis attending Hepatobiliary Ultrasound Clinic of the Hepatology Unit at Campus Bio-Medico University Hospital for regular surveillance program for hepatocellular carcinoma in the first 6 months of the study (from February to August 2017) were asked to participate to the study. In absence of liver histology, we referred to the clinical diagnostic criteria of liver cirrhosis. Specifically, we considered all cirrhotic patients with a history of long-lasting liver disease with different etiologies and morphological changes in the liver volume and structure evaluated during abdominal ultrasound and/or signs of PH: (1) hypertrophy of the caudate lobe/left lobe of the liver; (2) nodularity of the liver surface in the left lobe evaluated with linear probe; (3) coarse nodular pattern of the liver parenchyma; (4) portosystemic collaterals such as patent umbilical vein; (5) splenomegaly; and (6) ascites.

Endoscopic evaluation and grading of esophageal varices (OEVs) were performed within 3 months of the recruitment by a single expert endoscopist (>1000 examinations) who was blinded to the ARFI elastography results. Varices were classified as F1 (straight and small-caliber varices), F2 (tortuous veins forming bead-like appearance), or F3 (tumor-shaped varices) [15]. HRVs were defined as F2 to F3 or F1 with red wale marks, according to the Baveno V criteria [16].

The exclusion criteria were splenectomy, myeloproliferative syndromes, EGD older than 3 months from the enrolment, bad acoustic window for the liver and the spleen, severe ascites, inability of the patient to hold their breath, and refusal to informal consent.

pSWE was performed with Acuson S3000 ultrasound system (Siemens Medical Solutions, Mountain View, CA, USA) using a 1–4 MHz curved array probe and the software named Virtual touch quantification (VTQ^®^). All pSWE measures (ARFI elastography) were conducted by two sonographers blinded to the patient’s clinical data. First, a conventional B-mode US imaging study was conducted, which included the evaluation of liver parenchyma and volumes, spleen interpolar diameter, and its section area taken in a left longitudinal scan. During the ultrasound exam, the patients were asked to hold their breath in the supine position with the arms extended above the head and ARFI measurements, identified by a region of interest (ROI), were started in the liver and in the spleen. The ROI was characterized by a box with a fixed size of 10 mm × 5 mm, far from cysts, biliary ducts, principal blood vessels, or liver/spleen lesions. The ROI was placed in the liver parenchyma at the sixth and seventh liver segment two centimeters from the Glisson’s capsule and perpendicular to it using an intercostal ultrasound scan between the eighth and ninth right ribs. Afterward, the ROI was placed in the center of the spleen parenchyma, ideally two centimeters from the capsule and far from the vascular hilum, using an intercostal ultrasound scan between the eighth and ninth left ribs. The ARFI measurements were expressed in m/s as median and mean values. ARFI measurement failure was defined as zero valid shots, and unreliable measurements were reported as an interquartile range (IQR) to a median value ratio of >30% or a success rate of <60%. We performed the following ARFI ratio scores at the enrolment:

ALSDP (ARFI Liver–Spleen Diameter-to-Platelet Ratio Score): liver stiffness (m/s) × spleen diameter (mm)/platelet count (10^3^/mm^3^).

ASSDP (ARFI Spleen–Spleen Diameter-to-Platelet Ratio Score): spleen stiffness (m/s) × spleen diameter (mm)/platelet count (10^3^/mm^3^).

ASSAP (ARFI Spleen–Spleen Area-to-Platelet Ratio Score): spleen stiffness (m/s) × spleen area (cm^2^)/platelet count (10^3^/mm^3^).

ALSAP (ARFI Liver–Spleen Area-to-Platelet Ratio Score): liver stiffness (m/s) × spleen area (cm^2^)/platelet count (10^3^/mm^3^).

We also assessed whether patients met the following criteria:

Baveno VII criteria: liver stiffness of ≤20 kPa and platelet count of ≥150 × 10^3^/mm^3^ [17].

Expanded Baveno VI criteria: liver stiffness of ≤25 kPa and platelet count of ≥110 × 10^3^/mm^3^ [18].

Colecchia criteria: Baveno VI criteria (liver stiffness of ≤20 kPa and platelet count of ≥150 × 10^3^/mm^3^) and spleen stiffness of ≤45 kPa [19].

The study protocol was designed to provide 1 year of short-term follow-up (median: 14 months; range: 13–17 months) and 4 years of long-term follow-up (median: 46 months; range: 44–48 months). In the 2 scheduled follow-ups, we registered a laboratory assessment and the clinical patients’ evaluation, registering episodes of ascites, occurrence of new HRVs or OEV bleeding, and liver-related death.

The clinical investigation was conducted according to the principles described in the Declaration of Helsinki. The study was approved by the Ethical Committee of the University Hospital, and informed consent was obtained from all the patients due to the prospective nature of the study.

For descriptive statistics, categorical variables were shown as both number and proportion, while continuous variables were shown as a mean with standard deviation (SD) or a median with IQR, as appropriate. The association with incident liver-related events during follow-up was computed by means of Poisson regressions with robust error variance for binary data and expressed as adjusted risk ratios (aRR) with 95% confidence intervals (CI). Cross-sectional associations of ARFI ratio scores with actual liver-related events were estimated by means of logistic regression models and expressed as odds ratio (OR) with 95% CI. Multivariable models were corrected for age, sex, Child–Pugh score [20], and the previous presence of liver decompensation events.

Finally, the predictive capacities were verified through the AUC curve. A *p* < 0.05 was considered statistically significant. All analyses were conducted with R statistics 4.0.2 (R Foundation for Statistical Computing, Vienna, Austria).

## 3. Results

Among 114 enrolled patients, 14 were excluded for the following reasons: (1) no informed consent (*n* = 2), (2) bad acoustic windows (*n* = 2), (3) time between ARFI and endoscopy of >3 months (*n* = 5), (4) inability of patient to hold the breath (*n* = 1), and (5) severe ascites (*n* = 4). Finally, 100 subjects with liver cirrhosis (mean age: 66.7 years; women: 50%; Child–Pugh score A/B/C: 86%/12%/2%) were included in the study. A total of 99 and 82 patients were evaluated at the short- and long-term follow-ups, respectively (Figure 1). The most common etiologies of liver disease were chronic viral hepatitis (33%) and fatty liver (31%), which were present in more than half of the population. Overall, 10 (10%) subjects had previous OEV bleeding at baseline evaluation, while ascites, HRVs, and encephalopathy were present in 20 (20%), 20 (20%), and 9 (9%) patients, respectively (Table 1). In summary, the study population predominantly included patients with a well-compensated liver cirrhosis, namely, with a Child–Pugh score of liver function of ≤B7 and the absence of ascites or clinical evidence of encephalopathy, HVRs, and previous OEV bleeding at the enrolment. The small sample size of patients with decompensated liver cirrhosis did not exhibit statistical differences and significance in an advanced analysis.

Over a median of 14 months (short-term follow-up), 6 (6%) individuals died, while 6 (6%), 12 (12%), and 15 (15%) individuals experienced OEV bleeding, new onset of ascites, and new evidence of HRVs, respectively. One subject was lost at the short-term follow-up. After a median of 46 months (long-term follow-up), 17 (20%) individuals died, while 12 (15%), 33 (40%), and 26 (32%) individuals experienced OEV bleeding, new onset of ascites, and new evidence of HRVs, respectively (Table 2). Furthermore, 11 subjects were lost to the long-term follow-up.

Spleen ARFI, ASSDP, and ASSAP were significantly associated with HRVs in the prospective short- and long-term follow-ups and in the cross-sectional study (*p* < 0.05) (Table 3), while ALSDP and ALSAP were associated with HRVs only in the prospective long-term follow-up and the cross-sectional study (*p* < 0.05). All the ARFI ratio scores performed better than individual liver and spleen ARFI (AUC > 0.7) in detecting HRVs either at the enrolment or at the follow-up (Figure 2 and Figure 3), although ASSAP was the best ARFI ratio score for detecting HRVs at the long-term follow-up (AUC = 0.88) (Figure 4).

Furthermore, based on AUC values, ARFI ratio scores performed better than other validated combined models or criteria (such as Child–Pugh score [20], Baveno VII criteria [17], expanded Baveno VI criteria [18], and Colecchia criteria [19]) (AUC values between 0.5 and 0.6) in predicting the presence or the new occurrence of HRVs (Table 4). Moreover, only liver ARFI at the long-term follow-up (*p* = 0.039) and spleen ARFI at the short-term follow-up identified patients at risk of ascites (*p* = 0.009) in the prospective study. Other ARFI ratio scores were not associated with any other liver-related event, except for liver ARFI that was slightly associated with OEV bleeding (*p* = 0.057) during the short-term follow-up but not during the long-term period (*p* = 0.176) in the prospective study (Table 4).

## 4. Discussion

Over the past decades, many efforts have been directed toward the research concerning the assessment of the degree of PH with non-invasive tests [3,21] to rule out CSPH and specifically HRVs, reducing or avoiding an invasive diagnosis through HVPG and EGD, respectively, and also stratifying the risk of liver decompensation. Therefore, many authors have emphasized the role of LSM as a predictor of PH, showing a linear correlation between HVPG and LSM [7]. The Baveno VII Consensus Conference suggested that in patients with compensated advanced chronic liver disease, an LSM of ≤20 kPa and a platelet count of ≥150 × 10^3^/mm3 can identify the group of patients who most likely do not have HRVs (<5%) and for whom endoscopy can be avoided [17]. A possible limitation of the Baveno criteria is related to a substantially low number of spared endoscopies (15–25%) [22]. Therefore, most patients not presenting with HRVs still require endoscopy according to guidelines, and it is becoming evident that more accurate non-invasive scores are needed to improve risk stratification. To increase the number of spared endoscopies, Augustin et al. suggested the expanded Baveno VI criteria, a combination of an LSM of ≤25 KPa and a platelet count of ≥110 × 10^3^/mm [18].

Spleen stiffness measurement (SSM) by TE or pSWE has showed similar or even better accuracy versus LSM to identify patients at high or low risk of developing HRVs. The SSM cut-off of ≤46 kPa by TE has shown better specificity, with the same high sensitivity as the Baveno VI criteria in ruling out patients with HRVs [23]. In this setting, Colecchia et al. proposed the “Colecchia criteria”, a non-invasive prediction model that combined the Baveno VI criteria and an SSM of ≤46 kPa, avoiding more endoscopies than the Baveno VI criteria or an SSM of ≤46 kPa when considered individually [19]. The literature data showed that SSM might be a better marker of PH than LSM in patients with cirrhosis of viral etiology, and a growing body of evidence suggests that this might also be true for cirrhosis related to other causes [24]. We showed in a past preliminary study published as an abstract [25] that spleen ARFI was not useful for detecting HRVs in a cirrhotic population, while in the multivariate analysis, LSM (OR: 2.4 [1.17–5.34]), platelet count (OR: 0.98 [0.97–0.99]), and spleen interpolar diameter (OR: 1.16 [0.96–1.36]) were associated with HRVs. In contrast, Jain et al., in a cross-sectional study including 90 patients with liver cirrhosis, identified the best cut-off value for liver ARFI of 2.16 m/sec and of 3.04 m/s for spleen ARFI, in detecting OEVs, with a 92% of accuracy, when the stiffness values were taken together [26]. In a meta-analysis of 16 studies, SSM was able to detect GEVs with higher sensitivity and specificity (0.88 and 0.78, respectively) than LSM (0.83 and 0.66, respectively) [27]. These data are in line with our results, that showed SSM performing better than LSM in predicting HRVs at the short-term follow-up of the prospective study, with a risk ratio of 2.06 [(CI 95%: 1.06–4.03), *p* = 0.034]. Similarly, SSM showed an OR of 3.04 [(CI 95%: 1.23–8.65) *p* = 0.024] for the presence of HRVs in the cross-sectional study. In a recent meta-analysis of data from 45 studies about the ratio scores combining SSM and LSM with laboratory tests and spleen dimension, SSM and a ratio score of LSM by TE (kPa) × spleen size (in cm)/platelet count (number/mm^3^) [LSPS] were superior to LSM for GEV detection, with higher sensitivity (0.90 and 0.91 vs. 0.85), specificity (0.73 and 0.76 vs. 0.64), and AUC (0.899 and 0.851 vs. 0.817) [28].

Interestingly, Park et al. developed a novel ARFI-based prediction model that can accurately identify HRVs in patients with compensated cirrhosis: the ARFI Spleen–Spleen diameter-to-Platelet Ratio Score (ASPS), which was calculated as spleen ARFI × spleen diameter/platelet count. To detect HRVs, a negative predictive value of 98.3% was achieved at ASPS of <2.83, ruling out the presence of HRVs, whereas a positive predictive value of 100% was achieved at ASPS of >5.28 [29].

Nowadays, few studies have explored the role of ARFI in predicting HRVs in cirrhotic patients. Huang et al. [30], in an extensive study on 741 consecutive patients, combined liver and spleen ARFI measurements with platelet count, in a strategy named ARP, and compared them to the Baveno VI criteria [22] for OEV screening. In the training cohort, ARP strategy was defined as an LSM of <1.805 m/s or an SSM of <2.445 m/s and a platelet count of >110 × 10^9^/L, sparing 40.6% of EGDs with a missing rate of HRVs of 3.4%. In the validation cohort, ARP strategy improved the Baveno VI criteria avoiding unnecessary EGDs [30]. In another study, Wang et al. prospectively evaluated the performance of liver and spleen ARFI stiffness in a subpopulation of liver cirrhotic patients suffering from chronic hepatitis B under pharmacological viral suppression. The most accurate liver and spleen ARFI cut-offs were 1.46 m/s and 2.28 m/s, respectively. Combining these data with the platelet count (>150 × 10^9^/L) enhanced the capacity of the strategy to spare 38.6% of EGDs, with a low rate of missed HRVs in the validation cohort (3.4%) [31].

Our study demonstrated, in a cohort of patients with prevalent well-compensated cirrhosis (only 14% of the patients showed an impaired liver function identified by a Child–Pugh score of ≥B7), a higher diagnostic performance of ARFI ratio scores with respect to spleen and liver ARFI measurements considered individually for detecting esophageal HRVs, both at the time of enrolment and in the short- and long-term follow-ups. The relative bias related to the study population could be due to the characteristics of patients enrolled according to study design, undergoing regular ultrasound surveillance for hepatocellular carcinoma, predominantly as outpatients, in good clinical condition and without any overt clinical sign of CSPH. In particular, the presence of ascites or hepatic encephalopathy could negatively impact the ARFI performance because the reduced patient compliance and the ultrasound’s technical challenges. In the light of the small sample size of patients with decompensated cirrhosis, a potential impact of this sample size on statistical results has been observed.

In our study, the highest AUC test result at the long-term follow-up [0.88 (95% CI: 0.79–0.96)] was observed for ASSAP, even better than ASSDP [0.86 (0.78–0.95)], with the latter considering interpolar spleen diameter rather than spleen section area. One possible explanation for the better diagnostic performance of this ARFI ratio score could be related to the structural changes in the spleen and the volume secondary to PH [32], better identified by the measurements of the spleen section area than the interpolar spleen diameter in cirrhotic patients. Nevertheless, in our population, the use of ARFI ratio scores did not appear to be useful in identifying patients at risk of developing clinical events related to cirrhosis and PH, including death and OEV bleeding. Only spleen ARFI and liver ARFI were associated with the risk of developing ascites in the short- [(aRR (95% CI): 2.58 (1.27–5.24) *p* = 0.009] and long-term follow-ups [aRR (95% CI): 1.42 (1.02–1.98) *p* = 0.039)]. This is apparently in disagreement with the literature data, in which LSM is a validated predictor of clinical decompensation in patients with chronic liver disease [33], such that the current clinical guidelines of the European Association for the Study of the Liver (EASL) recommend the use of LSM to stratify the risk of clinical decompensation and mortality in compensated chronic liver diseases [34].

However, this study has some limitations. Firstly, the small sample size from a single center where the study population predominantly had well-compensated liver cirrhosis. This is due to the population of patients undergoing regular surveillance for hepatocellular carcinoma in the outpatient clinic, where the liver ultrasound is cost-effective for diagnosis and prognosis. However, this setting garnered higher interesting for exploration as, in this setting, the results showed a real clinical benefit for the patients. Indeed, once the CSPH is evident, some non-invasive tests used to predict HVRs could lose their clinical significance.

Secondly, since most of the viral cirrhosis was secondary to the eradication of hepatitis C virus with new direct-acting antivirals in recent years, a positive impact on PH progression, by reducing the onset of HRVs and clinical events related to liver cirrhosis, could be considered.

Finally, as stated before, the spleen measurement is highly sensitive but exhibits low specificity, with a high number of false positives in the case of splenomegaly due to non-hepatic causes.

Notably, as future directions, the model needs to be validated in a wider, external population, with a longer observation time.

## 5. Conclusions

Our study showed for the first time that combining liver and spleen ARFI measurements with spleen dimension and platelet count can provide some ratio scores capable of identifying well-compensated cirrhotic patients at risk of developing HRVs in the short- and long-term follow-ups. Particularly, the spleen section area seems to be of higher diagnostic relevance than interpolar diameter; thus, we recommend considering this parameter during the regular ultrasound surveillance of cirrhotic patients. We need further investigations and longer follow-up time to include this model in the clinical practice for selecting cirrhotic patients needing EGD but, above all, to inform patients and clinical hepatologists about the risk of HRVs, in order to start pharmacological or endoscopic prophylaxis when necessary. Finally, this is a good proof of concept in the field of multiparametric ultrasound of cirrhotic patients, and it should be adopted in the ultrasound room of each liver unit for a faster and more accurate clinical decision making.

## Figures and Tables

**Figure 1 diagnostics-14-00685-f001:**
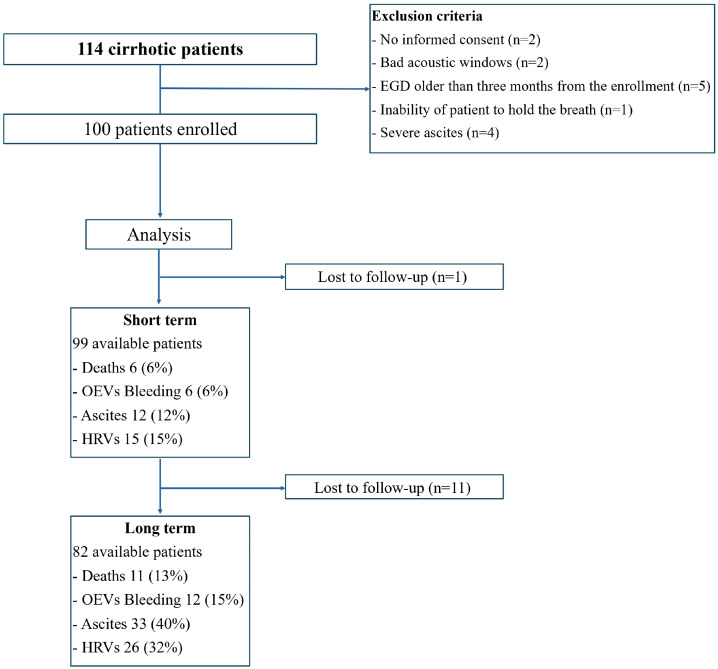
Design of the study.

**Figure 2 diagnostics-14-00685-f002:**
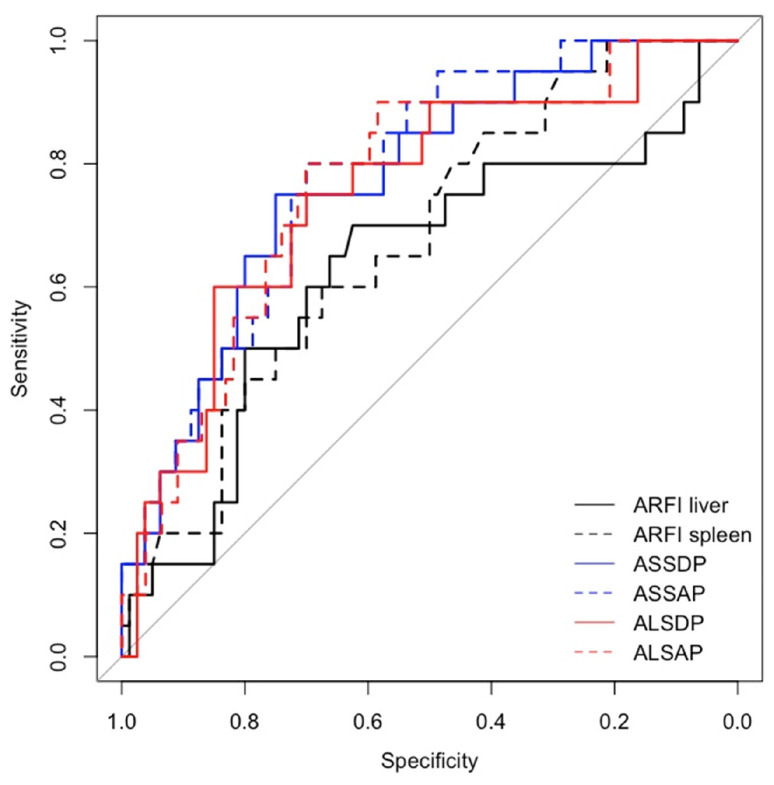
AUC for ARFI ratio scores for identifying HRVs.

**Figure 3 diagnostics-14-00685-f003:**
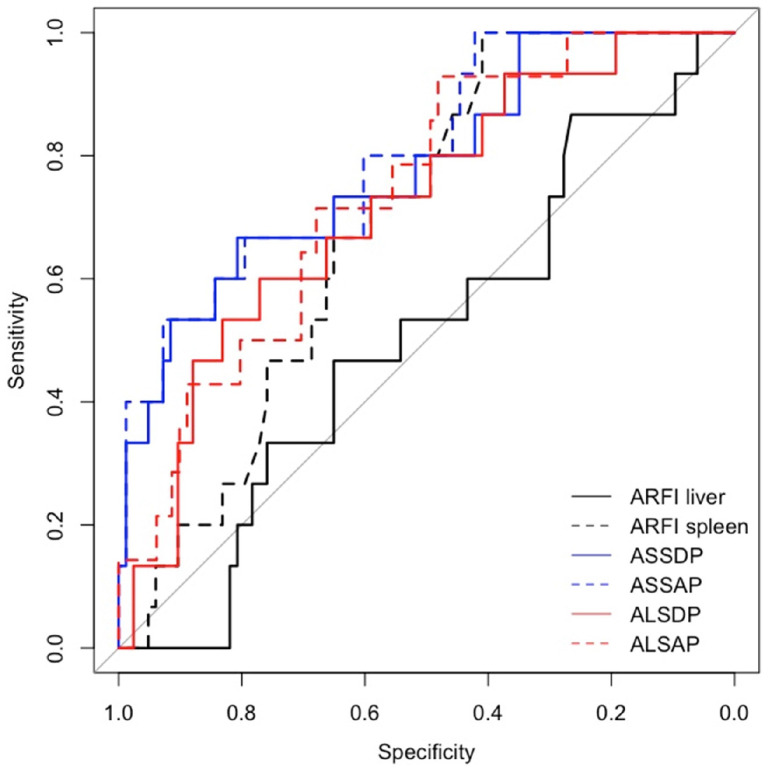
AUC for ARFI ratio scores for identifying HRVs at the short-term follow-up.

**Figure 4 diagnostics-14-00685-f004:**
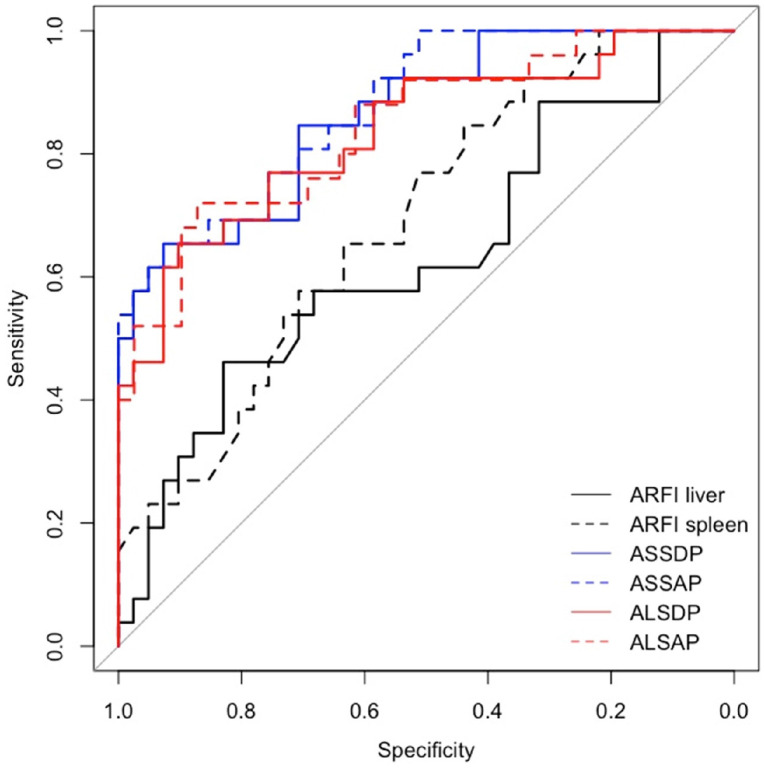
AUC for ARFI ratio scores for identifying HRVs at the long-term follow-up.

**Table 1 diagnostics-14-00685-t001:** Characteristics of the study participants and ARFI combined ratio scores at the enrolment.

N	100
Age (years), mean (SD)	66.7 (±10.5)
Female, *n* (%)	50 (50%)
Etiology	
Viral, *n* (%)	33 (33%)
Metabolic, *n* (%)	31 (31%)
Alcoholic, *n* (%)	17 (17%)
Other, *n* (%)	19 (19%)
Child–Pugh score, *n* (%)	
A	86 (86%)
B	12 (12%)
C	2 (2%)
Previous OEV bleeding, *n* (%)	10 (10%)
HRVs, *n* (%)	20 (20%)
Ascites, *n* (%)	20 (20%)
Encephalopathy, *n* (%)	9 (9%)
β-blockers therapy, *n* (%)	39 (39%)
Hepatocellular carcinoma, *n* (%)	5 (5%)
ARFI liver, median (IQR)	2.1 m/s (1.7–2.8)
ARFI spleen, median (IQR)	3.3 m/s (3–3.7)
ASSDP, median (IQR)	4.1 (2.4–6.7)
ASSAP, median (IQR)	17.1 (9.3–35.4)
ALSDP, median (IQR)	2.5 (1.4–5.2)
ALSAP, median (IQR)	10.9 (5.7–30.2)

Results are expressed as number (*n*) and proportion (%) as well as mean (SD) and median (IQR).

**Table 2 diagnostics-14-00685-t002:** Liver-related events at the follow-up.

**N**	**99**
Short-term follow-up, median (IQR) (in months)	14 (13–17)
Deaths, *n* (%)	6 (6%)
OEV bleeding, *n* (%)	6 (6%)
Ascites, *n* (%)	12 (12%)
HRVs, *n* (%)	15 (15%)
**N**	**82**
Long-term follow-up, median (IQR) (in months)	46 (44–48)
Deaths, *n* (%)	11 (13%)
OEV bleeding, *n* (%)	12 (15%)
Ascites, *n* (%)	33 (40%)
HRVs, *n* (%)	26 (32%)

Variables are shown as number (*n*), proportion (%), and median (IQR).

**Table 3 diagnostics-14-00685-t003:** Association of ARFI combined ratio scores with liver-related events.

	Prospective Study		Cross-Sectional Study
	Short-Term Follow-Up	Long-Term Follow-Up	
	aRR (95% CI), *p*	aRR (95% CI), *p*	aOR (95% CI), *p*
**Death**			
ARFI liver	0.89 (0.41–1.93), 0.772	1.35 (0.81–2.26), 0.248	-
ARFI spleen	1.65 (0.71–3.88), 0.247	1.37 (0.68–2.77), 0.378	-
ASSDP	0.99 (0.83–1.19), 0.911	1.04 (0.97–1.12), 0.264	-
ASSAP	1.00 (0.98–1.02), 0.965	1.00 (0.99–1.01), 0.423	-
ALSDP	0.91 (0.70–1.19), 0.511	1.05 (0.97–1.15), 0.240	-
ALSAP	0.99 (0.95–1.03), 0.629	1.01 (0.99–1.02), 0.439	-
**Ascites**			
ARFI liver	1.17 (0.62–2.22), 0.622	1.42 (1.02–1.98), 0.039	-
ARFI spleen	2.58 (1.27–5.24), 0.009	1.07 (0.68–1.67), 0.776	-
ASSDP	1.01 (0.9–1.13), 0.913	1.03 (0.98–1.08), 0.231	-
ASSAP	1 (0.98–1.01), 0.837	1 (1–1.01), 0.638	-
ALSDP	0.95 (0.84–1.08), 0.463	1.05 (0.98–1.12), 0.176	-
ALSAP	0.99 (0.97–1.01), 0.3	1 (0.99–1.01), 0.597	-
**HRVs**			
ARFI liver	0.73 (0.45–1.19), 0.204	1.11 (0.79–1.56), 0.561	1.65 (0.9–3.1), 0.111
ARFI spleen	2.06 (1.06–4.03), 0.034	1.81 (1.11–2.96), 0.018	3.04 (1.23–8.64), 0.024
ASSDP	1.12 (1.02–1.22), 0.016	1.09 (1.04–1.15), <0.001	1.22 (1.09–1.39), 0.001
ASSAP	1.01 (1–1.02), 0.014	1.01 (1–1.02), 0.001	1.03 (1.01–1.05), 0.002
ALSDP	1.07 (0.97–1.18), 0.172	1.13 (1.05–1.22), 0.002	1.21 (1.06–1.4), 0.005
ALSAP	1.01 (1–1.02), 0.165	1.01 (1–1.02), 0.007	1.03 (1.01–1.05), 0.004
**OEV Bleeding**			
ARFI liver	0.43 (0.18–1.03), 0.057	0.6 (0.29–1.25), 0.176	-
ARFI spleen	0.95 (0.38–2.35), 0.907	1.65 (0.71–3.84), 0.244	-
ASSDP	1.05 (0.91–1.21), 0.477	1.03 (0.93–1.13), 0.612	-
ASSAP	1.01 (0.99–1.02), 0.411	1 (0.99–1.02), 0.606	-
ALSDP	0.99 (0.85–1.14), 0.843	0.96 (0.84–1.11), 0.611	-
ALSAP	1 (0.98–1.02), 0.875	1 (0.98–1.02), 0.732	-

Prospective results are expressed as aRR with 95% CI. Cross-sectional associations are expressed as OR with 95% CI. All models were corrected for age, sex, and Child–Pugh score. Models for OEV bleeding were additionally corrected for previous OEV bleeding and the presence of HRVs. Models for ascites were additionally corrected for previous ascites.

**Table 4 diagnostics-14-00685-t004:** Diagnostic value of ARFI ratio scores for the presence or occurrence of HRVs in the prospective study.

	T 0	Short-Term Follow-Up	Long-Term Follow-Up
	AUC (CI 95%)	AUC (95% CI)	AUC (95% CI)
ARFI liver	0.63 (0.48–0.78)	0.5 (0.35–0.66)	0.63 (0.49–0.77)
ARFI spleen	0.67 (0.55–0.8)	0.69 (0.57–0.81)	0.69 (0.56–0.82)
ASSDP	0.77 (0.66–0.88)	0.78 (0.65–0.91)	0.86 (0.78–0.95)
ASSAP	0.78 (0.68–0.89)	0.8 (0.68–0.92)	0.88 (0.79–0.96)
ALSDP	0.75 (0.63–0.87)	0.72 (0.58–0.86)	0.83 (0.73–0.94)
ALSAP	0.76 (0.65–0.88)	0.74 (0.61–0.87)	0.84 (0.74–0.94)
Child–Pugh score	0.53 (0.44–0.63)	0.58 (0.46–0.69)	0.58 (0.49–0.67)
Baveno VII criteria	0.61 (0.48–0.73)	0.5 (0.37–0.64)	0.59 (0.48–0.7)
Expanded Baveno VI criteria	0.61 (0.5–0.73)	0.52 (0.41–0.63)	0.58 (0.49–0.67)
Colecchia criteria	0.54 (0.47–0.61)	0.48 (0.45–0.5)	0.54 (0.49–0.59)

## Data Availability

The data presented in this study are available upon specific request from the corresponding author (g.galati@policlinicocampus.it).

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
