# Peer review of "Acoustic Radiation Forced Impulse of the Liver and the Spleen, Combined with Spleen Dimension and Platelet Count in New Ratio Scores, Identifies High-Risk Esophageal Varices in Well-Compensated Cirrhotic Patients"

_diagnostics, 2024, doi:10.3390/diagnostics14070685_

Round 1
Reviewer 1 Report
Comments and Suggestions for Authors
The manuscript is very well written and of clinical and academic interest.
Please adjust the number of lost to follow up patients in table 1.
Other than that, the manuscript is accepted for publication.
Author Response
The Figure 1 has been reviewed according to the suggestion, so that the patient lost at the short-term follow-up was only one, despite at the long-term follow-up the patients were twelve.
Reviewer 2 Report
Comments and Suggestions for Authors
This prospective, single-center observational study, initiated in February 2017, included patients with cirrhosis. Esophagogastroduodenoscopy was performed within 3 months to assess esophageal varices (grading F1-F3) with high-risk varices (HRVs). pSWE (ARFI) measurements based on specific criteria were implemented. The study also introduced ARFI ratio scores (ALSDP, ASSDP, ASSAP, ALSAP) incorporating ARFI-based stiffness, spleen size/area, and platelet count. Patients were followed up on for a median of 14 months (short-term) and 46 months (long-term), monitoring for ascites, new HRVs, esophageal bleeding, and liver-related death.
Minor concerns:
•Typo errors: quite a few typographical errors—most of them are spacing errors were spotted.
•Limitations: please clarify some more limitations identified, such as the study population being restricted to well-compensated cirrhotic patients or the lack of long-term follow-up. In addition, in-depth mechanistic analysis and discussions are currently not explicitly available.
•Impact on results: the authors should discuss how these limitations might have affected the results. For example, they might try addressing that the results might not be generalizable to patients with decompensated cirrhosis.
•Future research directions: the authors may suggest some more insightful future research directions. For example, this might involve studies with a broader patient population or longer follow-up periods.
Comments on the Quality of English Languagenone.
Author Response
Minor concerns:
- Typo errors: quite a few typographical errors—most of them are spacing errors were spotted.
Authors’ reply: The entire text has been extensively reviewed and some typographical errors have been corrected.
- Limitations: please clarify some more limitations identified, such as the study population being restricted to well-compensated cirrhotic patients or the lack of long-term follow-up. In addition, in-depth mechanistic analysis and discussions are currently not explicitly available.
Authors’ reply: The limitations of the study have been more explicated both in results and discussion. Moreover, for the same reasons, we have modified the title as following: “ARFI of the Liver and of the Spleen, Combined with Spleen Dimension and Platelet Count in New Ratio Scores, Identifies High-Risk Oesophageal Varices in well-compensated cirrhotic patients”. This bias could be explained because the population studied was predominantly made by outpatients in regular surveillance for hepatocellular carcinoma by ultrasound.
In our study, the population of patients with a decompensated liver cirrhosis, that in other words could be patients with a Child Pugh score more than B7, or with a clinical manifestation of decompensation (ascites, encephalopathy, gastro-intestinal bleeding) already at the enrolment was too small to have a statistical weight, so that a deeper analysis was not needed.
- Impact on results: the authors should discuss how these limitations might have affected the results. For example, they might try addressing that the results might not be generalizable to patients with decompensated cirrhosis.
Authors’ reply: The concept that the results could be applied above all to a population of well compensated cirrhotic patients has been underlined. Indeed, it is in this population that the predictive role of non-invasive test such as ARFI combined with spleen dimension and platelets could change the clinical management. Once the patient experienced a decompensation (as bleeding, ascites, encephalopathy etc.), we well know that a clinically significant portal hypertension will drive the clinical events and it will affect the whole prognosis.
- Future research directions: the authors may suggest some more insightful future research directions. For example, this might involve studies with a broader patient population or longer follow-up periods.
Authors’ reply: The future research directions have been more explained now in the text and conclusions.
Reviewer 3 Report
Comments and Suggestions for Authors
1. The aim of this study was to assess the diagnostic value of ARFI in detecting and predicting the risk of liver-related events. However, the authors only provided details on the presence or new occurrence of HRVs. While ARFI is valuable in identifying cirrhotic patients at risk of HRVs, the focus during follow-up should be on variceal bleeding and mortality, rather than the development of HRVs.
2. The cohort size in this study was relatively small compared to similar studies, which may have limited the predictive value of ARFI. This limitation should be acknowledged as a potential constraint of the study.
Comments on the Quality of English LanguageThe quality of English in this article is pretty good.
Author Response
- The aim of this study was to assess the diagnostic value of ARFI in detecting and predicting the risk of liver-related events. However, the authors only provided details on the presence or new occurrence of HRVs. While ARFI is valuable in identifying cirrhotic patients at risk of HRVs, the focus during follow-up should be on variceal bleeding and mortality, rather than the development of HRVs.
Authors’ reply: We now revised the aims of the study, and we focused about the presence of HVRs at the enrolment and about the onset of new HRVs at the follow-up.
- The cohort size in this study was relatively small compared to similar studies, which may have limited the predictive value of ARFI. This limitation should be acknowledged as a potential constraint of the study.
Authors’ reply:
We well know this limit of the study, and this is now more underlined in the discussion and conclusion.
Reviewer 4 Report
Comments and Suggestions for Authors
The article: “ARFI of the Liver and of the Spleen, Combined with Spleen Dimension and Platelet Count in New Ratio Scores, Identifies Cirrhotic Patients with High-Risk Oesophageal Varices” shows new non-invasive method to detect cirrhotic patient with higher risk of variceal bleeding.
The study was well designed. Follow-up periods are long enough. The results are clearly presented. The discussion is smooth with good background of literature.
I recommend the paper for publication
Comments on the Quality of English LanguageI approve quality of English
Author Response
We approved the final version of the paper and we thank the referee's positive comments.
Reviewer 5 Report
Comments and Suggestions for Authors
In this well-planned, prospective single-center cross-observational study, the authors showed that combining liver and spleen ARFI measurements with spleen dimension and platelet count, can provide some ratio scores able to identify cirrhotic patients at risk of high-risk esophageal varices in the short- and long-term period. It is well known that the "gold standard" for the diagnosis of clinically significant portal hypertension (CSPH) is hepatic venous pressure gradient (HVPG) measurement. It should be recognized that until now, HVPG measurement is possible only in specialized centers. In addition, the procedure invasiveness and the need for repeated measurements increases a risk of possible complications and raises costs. At the Baveno VII consensus workshop, criteria were established for the exclusion or identification of CSPH based on liver stiffness values (estimated by transient elastography) in combination with platelet count. Previously, it was found that the stiffness of the liver and spleen on ARFI imaging correlate with the values of HVPG and make it possible to distinguish between cirrhotic patients with and without CSPH and esophageal varices. In addition, the correlation of portal hypertension with spleen dimension and platelet count is known. The authors have provided us with a new ratio scores for assessing high-risk esophageal varices, which is suitable not only for the initial assessment of portal hypertension, but also for its monitoring.
Despite the obvious advantages of this manuscript, it has a number of disadvantages:
1) it is necessary to formulate the purpose of the study more clearly, since it does not coincide with the conclusions drawn;
2) the present study is limited by small number of patients, and inclusion of patients from a single institution;
3) when discussing the problem, one should be guided by the recommendations of Baveno VII consensus workshop.
Comments on the Quality of English LanguageNo Comments
Author Response
- It is necessary to formulate the purpose of the study more clearly, since it does not coincide with the conclusions drawn;
Authors’ reply: Now the aim of the study has been clarified and it coincides with the conclusions. The changes have been made in the abstract as well as in the main text.
- the present study is limited by small number of patients, and inclusion of patients from a single institution;
Authors’ reply: We well know the limitations of the study. However, this is a “proof of concept” to take in consideration the multiparametric liver ultrasound not only in the surveillance of HCC, but also in the diagnosis and follow-up of complications related to the clinically significant portal hypertension. Our results are well supported by a strong rationale, but we know the actuarial limitations of the study.
- when discussing the problem, one should be guided by the recommendations of Baveno VII consensus workshop.
Authors’ reply: We have changed the reference to the newest Baveno VII recommendations about portal hypertension. Moreover, the extended Baveno VI criteria and Colecchia criteria were confirmed in our multivariate analysis.
Round 2
Reviewer 2 Report
Comments and Suggestions for Authors
The manuscript is now well-revised.
Please replace "In like manner" in line No 89 with "Likewise".
Author Response
Reviewer comment
Please replace "In like manner" in line No 89 with "Likewise".
Authors' reply:
We have changed the text as suggested.
Reviewer 3 Report
Comments and Suggestions for Authors The comments are sufficiently answered.Author Response
Reviewer comments
The comments are sufficiently answered.
Authors' reply:
We thank for the effort to enhance quality of the paper and of the journal.